# Mixed-Methods Approach: Impact of Clinical Consenter Diversity on Clinical Trials Enrollment

**DOI:** 10.3390/cancers17061043

**Published:** 2025-03-20

**Authors:** Angelica Sanchez, Christina M. Vidal, Noé Rubén Chávez, Nikita Jinna, Jackelyn Alva-Ornelas, Vanessa Myriam Robles, Cristal Resto, Nancy Sanchez, Dana Aljaber, Margarita Monge, Alicia Ramirez, Angela Reyes, Ernest Martinez, Veronica C. Jones, Jerneja Tomsic, Kendrick A. Davis, Victoria L. Seewaldt

**Affiliations:** 1City of Hope Comprehensive Cancer Center, Duarte, CA 91010, USA; angsanchez@coh.org (A.S.); cvidal@coh.org (C.M.V.); niwright@coh.org (N.J.); jalvao@coh.org (J.A.-O.); vanessa-myriam_robles@med.unc.edu (V.M.R.); cresto@coh.org (C.R.); nancsanchez@coh.org (N.S.); dbaljaber@ucdavis.edu (D.A.); vjones@coh.org (V.C.J.); jtomsic@coh.org (J.T.); 2Department of Social Sciences and Humanities, Charles R. Drew University of Medicine and Science, Los Angeles, CA 90059, USA; noechavez@cdrewu.edu; 3Department of Psychiatry & Neuroscience, University of California Riverside School of Medicine, Riverside, CA 92521, USA; margarita.monge@medsch.ucr.edu (M.M.); angela.reyes2@medsch.ucr.edu (A.R.); 4Department of Biochemistry, University of California Riverside, Riverside, CA 92521, USA; arami160@ucr.edu (A.R.); ernestm@ucr.edu (E.M.)

**Keywords:** mixed methods, inclusion, diversity, clinical trials, biomarkers

## Abstract

Historically, clinical trials have shown a significant lack of participant diversity. Ultimately, this lack of diversity in clinical trial recruitment widens the gap in the quality of healthcare treatment, delivery, and efficacy across populations. This study aims to explore the socio-cultural consenter qualities that may influence an individual’s decision to participate in clinical trials. Our results indicate that a consenter’s race and ethnicity were deemed as important among Women-of-Color (WOC), yet not important for Northern European White (NE White) women. Our findings provide evidence that additional studies are needed to better understand the factors that may influence a participant’s decision to enroll in a clinical trial study and ultimately enhance future recruitment for clinical trials.

## 1. Introduction

Lack of diverse recruitment to clinical trials limits the generalizability of trial results [1,2,3]. At City of Hope, recruitment by our clinical research team to our non-therapeutic/non-interventional clinical trials very closely matched the racial and ethnic demographics of our City of Hope Duarte Breast Oncology Clinic and our City of Hope Comprehensive Cancer Center Catchment Area; we neither under- nor over-accrued any specific ethnic or racial group. Our representative trial accrual occurred in the absence of (1) subject incentives, (2) attempts to over-recruit individuals from specific races or ethnic groups, and (3) special training for our consenting team. The only reason we could identify for our proportional accrual was that the individuals giving consent for research subjects for our trials were broadly representative of our Southern Los Angeles communities. Consequently, we hypothesized that perhaps the reason we were accruing diverse individuals to our non-therapeutic trials was due to the diversity of individuals who were giving consent to our trial subjects.

The consenting team, or the Clinical Research Assistant (CRA), is often the person who has the greatest contact with the research participant [4]. Therefore, a trusting relationship between the potential participant and the CRA is critically important in efforts to be more inclusive in clinical trial enrollment. We hypothesized that the ability of the CRA to garner trust from a variety of communities would directly impact the diversity of the trial participants.

There are many high-impact publications highlighting the (1) lack of diverse accrual to clinical trials and (2) potential strategies to increase trial participant diversity [5,6,7]. In a review of the literature, however, we found that there was a dearth of studies that considered the impact of consenter diversity on diverse clinical trial accrual.

To test our hypothesis that diverse consenters could increase trial participant diversity, we designed a prospective population-based mixed-methods investigation to examine factors (including the role of the consenter) on accrual to three non-therapeutic, non-intervention clinical trials at City of Hope Comprehensive Cancer Center (COHCCC). Our investigation comprised two components, a survey questionnaire, and in-depth in-person interviews. We hypothesized that diverse trial subjects might be more likely to enroll in a clinical trial when given consent by someone who looked like them, spoke their language, and understood their culture. Here we report our mixed methods evaluation of the impact of the socio-cultural background of the consenter on the decision of a potential research subject to participate in a clinical trial.

## 2. Materials and Methods

### 2.1. Trial Overview and IRB Approval

This study included both a survey questionnaire (IRB# 18205, see Appendix A) as well as an in-depth in-person interview (IRB# 18395). Patients who were previously approached for participation in one of three City of Hope Comprehensive Cancer Center (COHCCC) tissue- and blood-only, non-therapeutic/non-interventional breast cancer trials (BCTs) (IRB# 15418, 17009, 17185, see Appendix A) were then eligible for participation in this mixed-methods analysis. All trials were reviewed by the COH Institutional Board in accordance with national, state, and institutional regulations. A summary of each of these three non-therapeutic/non-interventional trials is provided below in Table 1; the full trial consents and protocols for all 5 studies are found in Appendix A. Subjects were approached in the order that they presented in a clinic by our Clinical Research Assistant (CRA) consenters. Subject demographics are presented below in Table 2.

### 2.2. Survey Questionnaire

Irrespective of whether subjects agreed or declined participation in one of the three non-therapeutic/non-interventional trials, subjects were then approached by a different CRA who provided consent information to participate in the survey study (IRB# 18205, see Appendix A). The survey study included an anonymous 5-min survey questionnaire of personal attitudes toward clinical trial participation. The survey consisted of a 39-item questionnaire to evaluate the following: (1) the decision to participate in or decline the previous trial and (2) attitudes towards clinical research overall. The goal of the survey was to elicit the role of the CRA consenter in the patient’s decision to enroll or decline participation in our three non-therapeutic/non-interventional trials and more broadly, any clinical research study.

The survey questions were developed from the relevant research literature, focus group studies, and interview studies that explored participant attitudes toward the consenting process (Appendix A). All participants were provided an informational consent form in compliance with the COH Human Research Protections Office prior to the survey. The first set of questions asked the patient to rate how much they agreed with nine statements that pertained to the recent communication they had with the CRA consenter. This was conducted using a 7-point Likert scale from “strongly disagree” to “strongly agree”. The nine statements (Q1–9) were adapted from Jenkins et al., 2009 and 2013 [8,9], Overholser et al., 2007 [10], and Heiney et al., 2010 [11] and captured participants’ understanding of this study being voluntary, as well as feelings of trust, support, sensitivity, and empowerment from the CRA consenter.

The next set of questions (Q10–14) captured the patient’s past and present participation in clinical research [12,13,14]. These questions asked whether the patient agreed to participate in the non-therapeutic BCT and about previous invitations and participation in clinical research studies. This section also sought to determine the influence of the community on the decision to participate in clinical trials and referenced who the participant knew that had ever participated in a clinical research study.

The subsequent set of questions (Q15–22) queried the importance of general consenter characteristics when approached for enrollment in a clinical research study. Options included “not important at all”, “a little important”, “important”, and “very important”. These questions were adapted from Myles, Heller, and Heiney [11,15,16]. This set included questions about the importance of consenter race, language, and gender in addition to temperament. The last section of survey questions (Q23–34) addressed the motivating factors to participate in any clinical research study. These questions were also rated on a similar scale of “not important” to “very important”. Motivating factors included altruism, religion, monetary compensation, and time commitment [17]. The questionnaire ended (Q35–39) with a collection of demographic information including race, ethnicity, language, and education level.

### 2.3. Statistics

Counts and percentages were calculated for qualitative data. To test associations, Chi-squared and Fisher’s Exact tests were performed. For the importance scale, both “important” and “very important” were considered “important” with every other response being considered “not important”. For the 7-point Likert scale, the responses were categorized as either “agree” which included “agree” and “strongly agree” with all other responses considered “disagree”.

### 2.4. In-Person Interviews

In a separate study, the CRAs and participating clinicians identified eligible participants who were being seen for a standard-of-care appointment with a healthcare professional within an outpatient setting. The CRA invited the patient to participate and obtained informed consent for the interview study (IRB# 18395, see Appendix A). If the participant was available to conduct the interview the same day, then a private room was acquired to complete the interview. If the participant was not available to conduct the interview the same day, then the interview was scheduled on a day when the participant would need to return for a standard-of-care appointment at City of Hope.

Structured individual interviews were conducted by trained CRAs (see Table 3). Our CRAs were trained to communicate clearly and establish rapport with the individuals they interviewed. They were also trained to avoid unintentionally biasing the participants’ responses and ensure that the interviewees stayed focused on responding to each of the questions. Interviewees and interviewers were matched based on gender and race to optimize the levels of comfort and psychological safety to rapport, cultural sensitivity, and open/honest communication. There was no training in accruing diverse individuals to clinical trials or an attempt to over-sample individuals of a specific race or ethnicity. The interview script was designed to capture patients’ understanding and concerns about clinical research, reasons for agreeing or declining participation, and facilitators or barriers to participation. To increase rapport, the CRA who gave consent for the patient into IRB# 18395 (see Appendix A) was also the same CRA who conducted the interview. With the consent of the patient, all interviews were audio-recorded with a digital audio recorder to ensure an accurate record of the patient’s responses. No monetary funds were offered. Participants were reassured they had the right to decline participation at any time and could decline to answer any questions during the interview. After the interview was completed, interviewers documented post-interview notes and thoughts about the interview.

Audio recordings were transcribed and reviewed for accuracy. A team of three CRAs read all transcriptions and independently developed broad themes and subthemes. Two independent coders coded each transcript, extracted phrases or sentences that reflected the themes, and then reached a consensus. Once the team of coders achieved at least a 95% inter-coder agreement, a final list of quotes associated with each of the codes/themes within the code book was finalized. The actual names of all participants were changed to pseudonyms when listed in each transcription.

## 3. Results

### 3.1. Response to Survey Questionnaire

Women approached for IRB# 18205 (see Appendix A) had previously been asked for permission to enroll in one of three non-therapeutic/non-interventional trials (Table 1); these three trials aimed to collect breast tissue, blood, and demographic information. All the women who had been approached for enrollment in these three trials were eligible for our subsequent survey questionnaire assessing factors related to clinical trial enrollment.

Two hundred and five women were approached in the order they presented to the Duarte Breast Oncology Clinic at City of Hope. There was no attempt to over- or under-sample women based on race or ethnicity. Out of the 205 women approached, twenty-four (11.7%) women declined to participate in this survey. The self-reported race and ethnicity of the women who declined were the following: 13/24 Non-Hispanic White, 2/24 Asian, and 9/24 Hispanic/Latina.

Of the 181 participants who completed the survey questionnaire, 94 (52%) were NE White and 87 (48%) were WOC (of the total number of subjects, 26% were Hispanic/Latina White, 17.1% Asian, 4.4% Black, and 1.7% Indigenous). See Figure 1 below for the study flow.

The demographics of the individuals accrued to our three non-therapeutic/non-interventional trials closely reflect the (1) cancer demographics of our City of Hope Duarte Breast Oncology clinic (52% Non-Hispanic White, 26% Latino/Latina, 8% Black, 14% Asian and Pacific Islanders, and <1% Indigenous) and (2) cancer demographics of our City of Hope Comprehensive Cancer Center Catchment Area: (57% Non-Hispanic Whites; 24% Latina; 11% Asian and Pacific Islanders, 8% Black, <1% Indigenous). See Figure 2 below.

### 3.2. Northern European, Non-Hispanic, White (NE WHITES)

There were 91 NE White participants who participated in the written survey (91 consented, 3 declined). A summary of Written Survey Results is presented below in Figure 2. There was no difference between the group that enrolled and that which declined in the understanding that this study was voluntary (*p* = 1) and that there was the freedom to withdraw (*p* = 1). There was a statistically significant difference between the participants who enrolled and declined with the feeling that the consenter created an atmosphere of trust and support. Nighty-three percent (*n* = 85) of enrollers felt this in contrast to only 67% (*n* = 2) of decliners (*p* = 0.05). There was also a significant difference in the percentages of NE White participants who disagreed with the statement that they felt empowered by the consenter to make their own decision regarding participation (0% of decliners vs. 5% of enrollers, *p* = 0.02). There was no difference among the other factors between the NE White enrollers and decliners.

### 3.3. Women-of-Color (WOC)

Among the 87 WOC enrollers (82 enrolled; 5 declined) who agreed to participate in the written survey, 26% were Latina/Hispanic White, 17.1% Asian, 4.4% Black, and 1.7% Indigenous. Approximately half of the WOC spoke a language other than English. Among the WOC enrollers, both those who enrolled and those who declined one of the three non-therapeutic/non-interventional studies agreed that their participation was voluntary and that they had the freedom to withdraw. Similar to the NE White participants, there was a statistically significant difference in the feeling that the consenter created an atmosphere of trust and support. Ninety-four percent (*n* = 77) of the enrollers, versus 60% of the decliners (*n* = 3), agreed with that statement (*p* = 0.05). Additionally, like the NE White participants, there was a statistically significant difference in the disagreement with the statement that as eligible participants they felt empowered by the consenter to make their own decision regarding participation (4% of enrollers (*n* = 3) vs 40% of decliners (*n* = 2), *p* = 0.02).

### 3.4. NE WHITES VS WOC

We observed statistically significant differences between NE Whites and WOC on questions regarding the consenter’s characteristics and their influence on the decision to enroll (see Figure 3 below). No NE White trial subject, (0%, *n* = 0) reported that consenter race impacted their decision to enroll; in contrast, 11% (*n* = 9) of WOC trial subjects stated the consenter race was important (*p* = 0.0009).

Similarly, none of the NE White trial subjects rated “looking like people in my community” as important; in contrast, 12% (*n* = 10) of the WOC trial subjects rated this as an important factor influencing their participation in a clinical study (*p* = 0.0004).

The language and the gender of the consenter, while not significantly different across the racial groups, did approach significance. Consenter language was deemed important to 69% of NE White enrollers and 56% of WOC (*p* = 0.07); this was an unexpected finding and likely reflects the diversity of Los Angeles’s immigrant communities. The consenter’s gender was considered “Not important” across racial groups (93% among NE White enrollers, 85% in WOC enrollers, *p* = 0.08). Further analysis showed that 68% of NE White and 51% of WOC considered “consenter language” important. We were surprised at the high percentage of NE Whites who considered “consenter language” important; we feel that this reflects the fact that many NE White individuals in Los Angeles are recent immigrants from Eastern Europe, the Middle East, and Armenia.

Non-consenter factors were important in a woman’s decision to enroll. Statistically significant differences were found with respect to the ability to benefit someone in the future (important to 89% of NE White enrollers and 98% of WOC enrollers, *p* = 0.04) and personal religious beliefs (important to 15% of NE White enrollers and 41% of WOC enrollers, *p* = 0.0001) (below in Figure 3D). Notably, feeling overwhelmed with the medical diagnosis and treatment was deemed important to 51% of NE White enrollers compared to 71% of WOC enrollers (*p* = 0.007). Additionally, there was a significant difference between the NE White women and WOC who answered “Yes” to 8 out of the 24 survey questions (below in Figure 3). These questions captured information that was deemed as “important” by participants on the following three main categories: “past and present participation in clinical trials”, “importance of general consenter characteristics”, and “motivating factors to participate in clinical research”.

### 3.5. Themes for In-Person Interviews

We conducted in-person interviews to better understand the perspectives, understanding, concerns, and attitudes of female patients toward clinical trials. The qualitative data resulting from the in-person interviews were conducted to supplement the quantitative information collected via the survey study (IRB# 18205, see Appendix A). From August 2019 to February 2020, we approached 61 clinical trial subjects who were (1) previously given consent for one of these three non-therapeutic/non-interventional studies (IRBs 15418, 17009, 17185, see Appendix A) and/or (2) were within five years of their initial breast cancer diagnosis and/or (3) at high risk for breast cancer. Women were approached in the order they presented to the clinic. A total of 44 women agreed to be interviewed. The racial/ethnic makeup was the following: 20 Latinas ([White/Hispanic]-1 Salvadorian, 1 half Mexican/half Armenian, and 18 Mexican), 8 Asian (1 half Chinese/half Japanese, 1 Indian, 3 Chinese, 2 Vietnamese, 1 Thai), 8 African American/non-Hispanic, and 8 White/non-Hispanic (2 Iranian and 6 European). The age range of women was from 27 to 71 years of age. Seventeen women declined participation. The racial/ethnic makeup for the women who declined was the following: eight Asian, four Latinas, four African American/non-Hispanic, and one White/non-Hispanic. Demographics of the women participating in in-person interviews are presented below in Table 2; interview questions are presented below in Table 3.

The interviews were conducted by three CRAs and one postdoctoral fellow. We matched the race/ethnicity of the interviewer to that of the interviewee: (1) Asian CRAs interviewed participants who self-identified as Asian (both east- and west-Asian), (2) one White Latina CRA interviewed both White Latinas and Non-Hispanics, and (3) one African American postdoctoral fellow interviewed African American participants. All interviews were conducted in English and ranged from 4 to 49 min in length.

The themes identified included scientific advancement, altruism, feelings/attitudes toward clinical research studies, incentive, research literacy, barriers/facilitators, patient-centeredness, cultural responsiveness, and consenter qualities. These themes were represented in responses from all races/ethnicities.

*Positive:* Common throughout the themes was a desire to advance the scientific field and a desire to help others. One patient (African American) stated, “if I knew that my contribution is going to not only improve the chances of them finding a better treatment but also that everyone would have access to”. Another patient (Asian) commented, “in health care too I guess each clinical research is different, but each research protocol has a different purpose in terms of advancing medical treatments possibly finding a cure, helping you know. I guess in the end, better medical treatments. More efficient and more efficacious.” The desire for scientific advancement was the most common positive theme among each racial/ethnic group.

*Negative:* Negative feelings toward clinical research studies elucidated feelings of mistrust in the medical field. This was seen across races, but in the NE White patients, this was seen on a more individual level while in the WOC participants, this was expressed on a community level. One patient (NE White) stated, “if you get a placebo and you don’t know what it is and the doctors don’t know whatever. There should be risk there either um maybe even when you’re not getting a placebo could make you worse.” However, cultural mistrust was a sub-theme only found in the WOC participants. One patient (Asian) commented, “I certainly think that Asian people have a harder time trusting medical providers. I’m a medical provider so I don’t have that problem (laughs). But um...certainly I think my relatives certainly have a hesitation when it comes to trusting medicine in general, and then when you’re introducing clinical trials, I think they also have the fear of being treated like uh..I don’t know like an experiment.” Another (African American) stated, “I just think that in my culture, you know, our culture, I don’t think we’re taught to do that, we’re taught that what stays at home, stays at home.” Still another (Hispanic) commented, “I feel that, in our culture sometimes when they say clinical research is like I’m being, I’m a guinea pig, or they’re just going to experiment and, what if it doesn’t work and they’re just trying it on me.” African American patients also expressed the most concern about confidentiality.

*System barriers/facilitators:* All races/ethnicities commented on barriers to participation in clinical trials. The sub-theme presented across all groups included time/distance. One patient (Asian) commented, “one thing because I live really far from the hospital so if like let’s say today it happened to be on the same day with my appointment which is really nice so I don’t have to make a sixty minutes’ drive out here just to do the research that would definitely affected my uh decision but that’s the only thing is that if you can schedule the same day as my other appointments that that would be great.” Other sub-themes included research literacy and language barriers. Another (Asian) commented, “that’s my biggest gripe is sometimes we can’t consent people because there’s not an ICF in their native language and then they don’t qualify, right? Nor they can’t consent. Um…so that’s a challenge.” Language barriers were commented on more often among the Asian participants than any other race/ethnicity.

*Consenter qualities:* All races and ethnicities also commented on the consenter qualities as being important to participation in research. Sub-themes included professionalism, communication skills, and emotional sensitivity. Patients also commented on the legality of the consent forms and the barrier introduced by the number of pages in the consent form as well as the number of signatures required. Across all populations, patients wanted the consenter to explain the study in a thorough yet concise way.

## 4. Discussion

Diverse accrual to clinical trials is essential to ensure that trial results are generalizable, and drugs are safe and effective for all people [18]. Accordingly, the NCI evaluation of diversity in clinical trials is part of the metric used to assign accreditation to cancer centers. Despite these efforts, there is a marked underrepresentation of individuals who self-identify as Latina/Hispanic, Asian, and African-American/Black/African in both non-therapeutic and therapeutic clinical trials [19,20,21].

Here we accrued individuals to three trials that collected tissue, blood, and demographics. Our 90% accrual rate is consistent with other tissue studies but contrasts with other tissue studies that frequently under-represent Hispanic/Latino, Black/African American/African, Asian, and Indigenous individuals [22,23]. For example, in the 5729 samples collected for The Tissue Genome Atlas (TCGA) compared with the 2022 Census Data, the TCGA over-recruited Non-Hispanic Whites and under-recruited individuals who self-identified as Latino/Hispanic and Asian: Non-Hispanic White 58.9% (vs. 77% TCGA; 29% *over*-recruitment), Black 12.6% (vs. 12% TCGA; 5% under-recruitment), Asian 6.1% (vs. 3% TCGA; 51% under-recruitment), and Hispanic/Latino 19.1% (vs. 3% TCGA; 84% under-recruitment) [24]. The lack of inclusion in non-therapeutic studies such as the TCGA study and therapeutic trials, has hampered the generalizability of some study results.

There is a need to address the barriers to diversity of enrollment into both therapeutic and non-therapeutic trials [25,26,27]. Clinical trial participant diversity is a problem with many causes and no easy one-size-fits-all solution; multiple barriers exist to increasing diversity in clinical trials [28,29,30].

Unlike many studies, our recruitment to clinical trials closely matched our City of Hope Duarte clinic and Comprehensive Cancer Center Catchment Area demographics—neither under- nor over-representing any racial or ethnic group. We offered no incentives to participate; there were no efforts to single out one race/ethnicity over another; we offered no special training or direction to the individuals giving consent for clinical trials. Given a lack of any special directives or incentives, we hypothesized that perhaps the reason we were accruing diverse individuals to our non-therapeutic trials was due to the diversity of individuals who were giving consent to our trial subjects.

This mixed-methods study, including a survey questionnaire and in-person interviews, takes a holistic look at barriers and facilitators to increasing diversity in clinical trials. The survey questionnaire highlights the importance of consenter characteristics in WOC participants’ decision to enroll in clinical trials. For WOC participants, identifying with the racial/ethnic background of the consenter was an important factor in the decision to enroll in the clinical study. Our findings support the barriers identified in other studies. Regnante et al. investigated key practices that resulted in minority accrual between 10 and 50% onto clinical trials and found that intentionality in race reporting and documentation, the barriers to participation and focused strategies to overcome those, and the inclusion of the practitioner in promotion of the clinical trial were essential [31].

There were differences seen in the motivating and inhibiting themes identified among the NE White and WOC participants. The in-person interviews highlighted feelings of cultural mistrust among the WOC participants but not in the NE Whites participating in this study. This is further reflected in the survey questionnaire rating the importance of the cultural background of the consenter. The combination of the results from both these methods highlights the influence of cultural mistrust as a barrier and the importance of the diversity of the clinical research team in overcoming this barrier, particularly among WOC clinical trial participants. Furthermore, there is heterogeneity within the WOC population; each community is influenced by its own historical and present contexts that influence its relationship with the medical system. Approaches to overcome this barrier must address the specific needs of each community.

Altruistic motivations, such as benefiting others were more important for WOC enrollers than NE White enrollers. Personal religious beliefs were considered more important for WOC enrollers than NE White enrollers in motivating participation. For many racial/ethnic minority communities, religion is central to their culture, values, views of life and death, sense of community, health, and well-being overall, and therefore it would be a factor that would understandably frame and guide decision-making in the context of a clinical trial that has implications for health, well-being, and life and death considerations.

WOC enrollers reported a stronger impact of “feeling overwhelmed with the medical diagnosis and treatment” as a factor for decision-making; this may speak to the need to feel greater support with the challenge of navigating and overcoming overwhelming and difficult emotions tied to their medical diagnosis and treatment. These specific findings call for more exploration of religious beliefs and emotional or mental health challenges faced by racial/ethnic minorities in their medical care journey and their seeking of appropriate support, empathy, and trust as they are navigating and making critical decisions for improving their health and participating in clinical trials that can offer help to them or other patients.

## 5. Study Limitations

This study captures immediate feedback regarding the decision to enroll or decline participation in a clinical research study, it has several important limitations that affect the generalizability of our findings. First, our non-therapeutic trials have high enrollment, which limits our ability to capture factors associated with declination. Secondly, we have a limited number of participants in both the survey questionnaire and in-person interview. Thirdly, as this study focuses on non-therapeutic trials, it does not capture factors related to therapeutic clinical trials. Fourthly, our study lacks a control arm. A potential confounder we did not consider was the relative ages of the consenters vs. the interview subjects. The consenters and interviewers were younger than the subjects; this could have either a positive or negative impact on accrual and/or our results. Moving forward, we will account for this potential confounder in future trials.

Other key issues are that our study is limited to a single institution, has a small sample size, is limited to women only, and is limited to a specific geographic region (Los Angeles, California). Additional studies are required to validate our findings and test their generalizability in other populations.

## 6. Conclusions

Despite its limitations, our study provides evidence that a clinical research team can increase clinical trial diversity. Our study provides evidence that racial/ethnic and socio-cultural qualities may influence a participant’s decision to enroll in a clinical trial. However, additional studies are needed; these include research across age groups and sexes. In our non-therapeutic/non-interventional trials, we accrued trial subjects in the order that they presented to our clinic; recruitment was in line with our catchment cancer demographics. In order to conduct this study, it was important for all subjects to be accrued equally—without any special training or attempt to over-/under-accrue individuals of specific races or ethnicities. In our future studies, we aim to capture additional viewpoints through multi-institutional studies in communities that are different from our Los Angeles clinics (e.g., clinics that primarily serve African American/Black/African women). Generalizability, quality, and effectiveness of healthcare treatment require diverse representation in all clinical trials. In future directions, we will test the impact of consenter diversity on clinical trial accrual diversity in therapeutic trials that span multiple institutions and include both men and women.

## Figures and Tables

**Figure 1 cancers-17-01043-f001:**
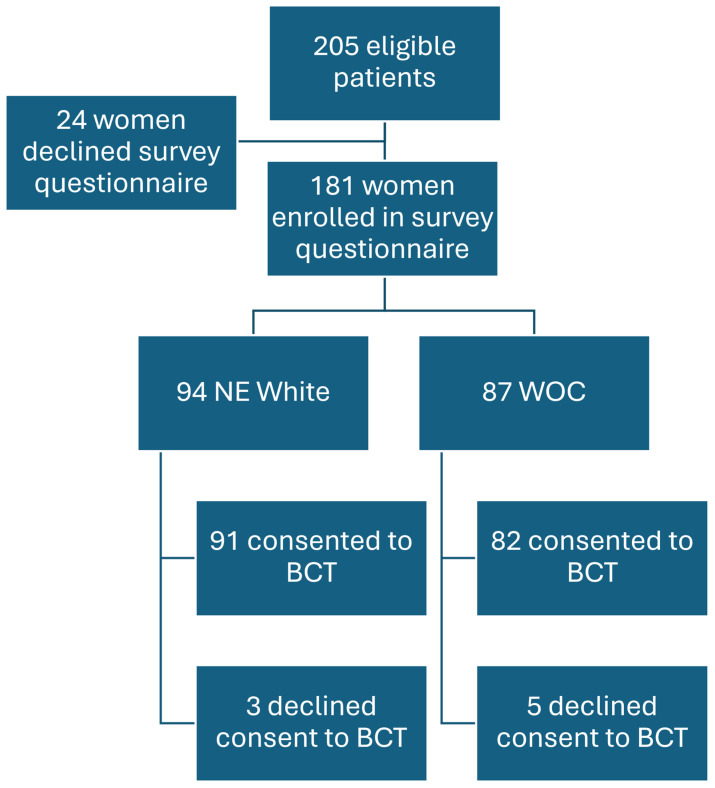
Flow diagram of eligible study participants. Schematic illustrating the breakdown of study participants. A total of 205 eligible participants were approached. A total of 181 women ultimately enrolled in the survey study: 94 of the women self-identified as NE White and 87 self-identified as WOC. Of the 94 NE White women who completed the survey, 91 consented to BCT, and 3 declined BCT participation. Of the 87 WOCs that completed the survey, 82 consented to BCT, and 5 declined BCT participation.

**Figure 2 cancers-17-01043-f002:**
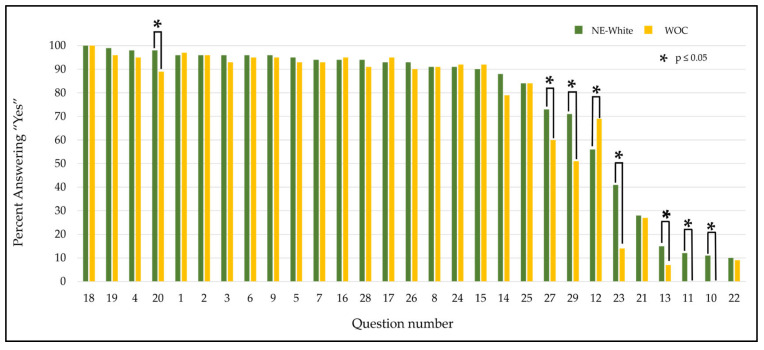
Diversity in “Yes”-answered survey questions. Percent of “Yes”-answered questions was calculated in both the NE White and WOC cohorts. WOC answered yes to a higher percentage of questions compared to NE White women, who only answered 1 out of 29 questions more than WOC. *n* = 181 * *p* ≤ 0.05.

**Figure 3 cancers-17-01043-f003:**
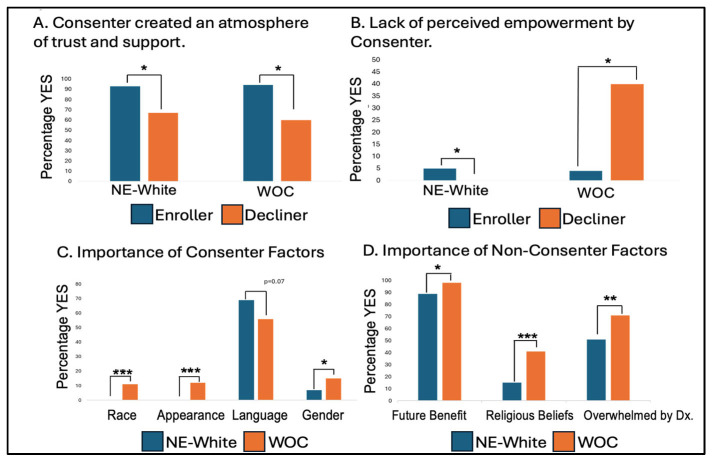
Differences in survey responses between NE White women and WOC. (**A**) Difference in the perceived trust and support created by the consenter in NE White women enrollers (*n* = 85) versus those who declined (*n* = 2) and WOC enrollers (*n* = 77) versus decliners (*n* = 3) * *p* = 0.05. (**B**) Comparison between NE White women enrollers (*n* = 5) and decliners (*n* = 0) and WOC enrollers (*n* = 3) and decliners (*n* = 2) on not feeling empowered by the consenter to make their own decision to participate * *p* = 0.02. (**C**) Differences in NE White women and WOC regarding the importance of consenter factors: race *** *p* = 0.0009, appearance *** *p* = 0.0004, language *p* = 0.07, and gender * *p* = 0.08. (**D**) Differences in NE White women and WOC regarding the importance of non-consenter factors: future benefits of research * *p* = 0.04, religious beliefs *** *p* = 0.0001, and feeling overwhelmed by diagnosis ** *p* = 0.007.

**Table 1 cancers-17-01043-t001:** Clinical trial information and data collected.

CoH IRB	Title and Purpose	Consent in Person (Y/N)	Consent in Writing (Y/N)	Data and Specimen Collection
**15418**	***Title***: Breast cancer initiation events in the high-risk population. ***Purpose***: Collect breast tissue and blood in individuals at >20% increased lifetime risk for or diagnosed with breast cancer.	Y	Y	Breast tissue, blood, DNA/RNA, demographics, cancer screening or breast cancer diagnosis data including treatment, imaging, outcomes, genetic testing, and medical information.
**17009**	***Title****:* Combined breast MRI/biomarker strategies to identify aggressive biology. ***Purpose***: Collect 3 extra core biopsy specimens from women at high-risk for or diagnosed with breast cancer who are undergoing MRI screening and require a core biopsy	Y	Y	
**17185**	***Title***: Breast microenvironment signaling during cancer initiation. ***Purpose***: Collect breast tissue and blood in individuals at >20% increased lifetime risk for or diagnosed with breast cancer.	Y	Y	

**Table 2 cancers-17-01043-t002:** Demographics of the women participating in in-person interviews.

Race	*n* (%)
Asian	8 (18%)
Black	8 (18%)
White	28 (63%)
**Ethnicity**	
Hispanic	21 (48%)
Non-Hispanic	23 (52%)
**Age**	
25–35	8 (18%)
36–45	6 (14%)
46–55	14 (32%)
56–65	11 (25%)
66–75	5 (11%)
**Dx of Breast Cancer in Past 5 Years**	
Yes	41 (93%)
No	3 (7%)
**US Born**	
Yes	25 (57%)
No	15 (34%)
Unknown	4 (9%)
**Marriage Status**	
Single	15 (34%)
Married	26 (59%)
Divorced	3 (7%)
**Religious**	
Yes	35 (80%)
No/Unknown	9 (20%)
**Highest Level of Education**	
Graduate or professional school	7 (16%)
College degree	13 (29.5%)
Some college or associates	8 (18%)
Vocational or technical school	4 (9%)
High school	7 (16%)
Some high school	2 (4.5%)
Unknown	3 (7%)

**Table 3 cancers-17-01043-t003:** In-Person Interview Questions.

Have you participated in clinical research before? Yes: What was the study about? No: Why not? What does clinical research mean to you? Probes: In your own words, what is clinical research? What comes to mind when you think of clinical research? How do you feel about clinical research? Probe: Do you have any strong feelings about clinical research? What concerns, if any, do you have about participating in clinical research? Probes: Do you see any potential risks? Can you think of any barriers that might make it difficult for *you* to participate in clinical research? Do you think it is challenging for people in your community or cultural/ethnic group to participate in clinical research? Probes: If yes, how is it challenging? If no, how is it not challenging? What are some reasons, in your opinion, for why people may want to participate in clinical research? Probes: Of the reasons that you mentioned, what are the most important? Were there any people at City of Hope who spoke to you about participating in clinical research? Probes: Who were some of these people? What did they tell you? Did they influence your decision to participate or not participate? A consenter is the person who tells you about the study and asks you if you want to participate. In general, what characteristics/qualities are essential for a consenter to have? Probes: What should consenters do to help people make their decision about participating in clinical research? What should institutions, like City of Hope, do to better inform patients about clinical research? What should institutions do to help engage more patients and others in participating in clinical research? Is there anything else we didn’t discuss about clinical research that you would like to share or think we should know?

## Data Availability

The data generated during and/or analyzed during the current study are available from the corresponding author upon reasonable request.

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
