# Peer review of "Mixed-Methods Approach: Impact of Clinical Consenter Diversity on Clinical Trials Enrollment"

_cancers, 2025, doi:10.3390/cancers17061043_

Round 1
Reviewer 1 Report
Comments and Suggestions for Authors
This is a very insightful and important study to understand the influence of the consenter’s race and ethnicity on the participant’s willingness to participate in clinical trials. The findings from the study will be influential to identify strategies to increase minority participation and diverse groups in clinical trials. A major strength of the study is the race/ethnicity of the consenters to match the ethnicity of the clinical trial participants to address an important question as to what extent does socio-cultural consenter qualities influence a patient’s decision to participate in clinical trials.
I only had 2 minor comments
In abstract, line 37 and results line 159: 205 women were approached of those 94 were NE-white and 87 were WOC. What was the ethnicity of the remaining 24 who did not agree to participate?
Did the study take into account age of the consenter compared to the participant? Could this be a confounder?
Author Response
Concern 1: In abstract, line 37 and results line 159: 205 women were approached of those 94 were NE-white and 87 were WOC. What was the ethnicity of the remaining 24 who did not agree to participate
Response 1: The race and ethnicity of the 24 women who declined self-identified as 13/24 Non-Hispanic White, 2/24 Asian and 9/24 who identified as Hispanic/Latina.
Concern 2: Did the study consider the age of the consenter compared to the participant? Could this be a confounder?
Response 2: This is a really good question and one that we did not consider. Our consenters were younger than the women they were consenting. The age relationship was that of mother (subject) to daughter (consenter). This age difference could influence consenting rates – either positively or negatively.
In the revised manuscript we discuss age as a variable that could 1) either positively or negatively impact study results and 2) could act as a confounder. We will consider this variable in our next study.
Reviewer 2 Report
Comments and Suggestions for Authors
This article is an analytical conclusion of a research report. In clinical research, ensuring that a technology or drug is effective for different populations is an important area of study, but it is indeed influenced by other factors. The author has analyzed aspects such as the race, ethnicity, and gender of the participants, and the research approach is good, but there are still some issues, which I will list as follows.
- The interview sample size of participants is small, with only 44 people, and the survey response group consists of only 181 people. Is this sample representative?
- As a research report conclusion, the focus should be on the conclusions drawn from data analysis, yet the inclusion of the complete survey questionnaire occupies a large portion of the report and is unnecessary. The questionnaire could be included in an appendix or supplementary materials.
- There is an imbalance in the representation of different ethnic groups. Can this imbalance be addressed by using other centers or methods to improve the balance of the analysis?
- The flowchart in Figure 1 is somewhat rough, with small and unclear text.
- It would be beneficial to present the research data and conclusions for different issues through visual analysis, such as charts or images.
- The Introduction section didn't label any references, and there is no adequate review of similar studies in this regard.
